# Experimental Dynamic Response of a Multi-Story Frame Structure Equipped with Non-Conventional TMD Implemented via Inter-Story Isolation

**Michela Basili** [1,*] and **Maurizio De Angelis** [2]

1   Department of Economics, Universitas Mercatorum, Piazza Mattei 10, 00186 Rome, Italy
2   Department of Structural and Geotechnical Engineering, Sapienza University of Rome, Via Eudossiana 18, 00184 Rome, Italy
*   Correspondence: michela.basili@unimercatorum.it

**Abstract:** A shaking table experiment conducted on a multi-degrees-of-freedom frame structure equipped with a non-conventional tuned mass damper (TMD) is presented. The non-conventional TMD is characterized by a high mass ratio, without adding further structural masses, and is realized via inter-story isolation. The structure top story mass of a four-story steel frame structure is isolated and converted into tuned mass, connecting to the substructure with two high damping rubber bearings placed in series. Aspects related to the dynamic structural response as well as the seismic effectiveness assessment of a non-conventional TMD are addressed. Three structural configurations are tested: the reference four-story structure, the three-story intermediate structure, and the three-story structure equipped with a non-conventional TMD. The input motion conditions considered are: white noise, sine sweep, and natural earthquakes. Through experiments, structural identification is carried out and different dynamic behaviors emerge for the configurations tested. The nonlinear effects provoked on the structure by the adopted isolators are investigated, showing high dissipative capabilities in a wide range of amplitudes of the excitation. It is demonstrated that a non-conventional TMD is a smart control strategy useful for enhancing structural vibration mitigation.

**Keywords:** passive control; non-conventional TMD; inter-story isolation; high damping rubber bearings; shaking table tests; experimental dynamic response; structural identification; nonlinear effects; earthquake excitation

## 1. Introduction

Among the passive control strategies proposed to attenuate structural vibrations due to dynamic inputs, the use of a passive tuned mass damper (TMD) has been largely studied in the past decades [1]. The control system consists of a small auxiliary mass attached to a main structure by means of a linear or nonlinear connection, tuned to reduce the primary structure oscillations (Figure 1a). A conventional TMD works effectively, especially when applied to reduce vibrations due to periodic excitation, but it seems not robust for structural parameter variations and fails when dealing with non-stationary inputs, especially of impulsive character.

Recent studies have demonstrated that TMD control performances and robustness can be strongly enhanced by increasing its mass [2,3]. Research has proven inter-story isolation to be an effective method to achieve passive control of structural vibrations. The idea of applying partial isolation for original structures dates back to some decades ago [4,5]. The technique, compatible with applications on new buildings as well as retrofit interventions, consists of the insertion of isolators in the columns of multi-story buildings at floor level, resulting in the creation of a non-conventional TMD system, characterized by a high mass ratio without adding further structural masses (Figure 1b,c). Compared to traditional TMD

devices, inter-story isolation ensures realization simplicity, minimum impact on target buildings, and contained construction and maintenance costs.

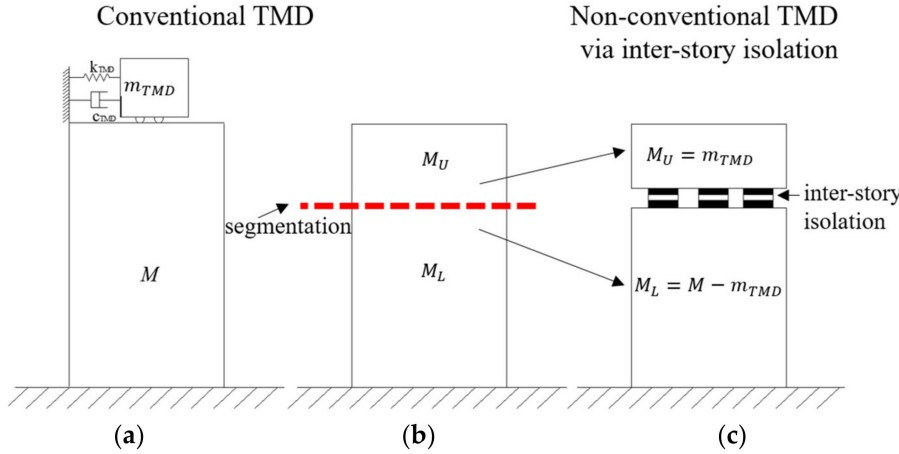

**Figure 1.** (**a**) Structure with conventional TMD, (**b**) segmented structure, (**c**) structure with non-conventional TMD implemented via inter-story isolation. Red dotted line indicates structural segmentation.

In the case of frame structures, a way to apply this concept is to use masses already present in the structure to be converted into tuned masses, realizing segmented upper stories or sliding roof systems, isolated from the substructure and acting as TMDs [3,6]. This idea is to provide supplemental damping by inducing a vibration energy transfer from the structural portion below the isolation system to the structural portion above it. Top, intermediate, and base discontinuity in frame structures to investigate the capability to mitigate the effects of seismic ground motions are discussed in [7,8]. Ref. [9] gives insights into inter-story isolation design applied to real buildings which possess different dynamic characteristics. In [10], seismic retrofitting of existing masonry buildings through an inter-story isolation system is proposed. The idea of isolating masses to be utilized for control purposes can be also achieved with semi-active devices [11]. A new alternative is to use an inerter assembled with a conventional TMD to join the advantages of a non-conventional TMD without utilizing a large physical mass amount, since this is furnished as inertial mass by the inerter itself [12–14]. Recently, Refs. [15,16] applied the concept of substructuring with a tuned-inerter-damper to enhance the performance of a mega substructure system.

Among the literature reports concerning applications of a non-conventional TMD, recently studies that focus numerically on the amplification of seismic demands in inter-story isolated buildings subjected to near fault pulse type ground motions [17,18] are reported. Few studies instead regard experimentations conducted on a non-conventional TMD applied for inter-story isolation [19–21]. In [19], they explored experimentally the application of an aseismic roof isolation system with a small-scale laboratory model. Ref. [20] proposed the utilization of a non-conventional TMD for application on an industrial steel structure, tested on a shaking table. The concept of a non-conventional TMD in a theoretical and experimental study on an innovative seismic retrofit solution realized adding new isolated upper stories on a reinforced old brick masonry building is applied in [21]. It is also worth mentioning the application implemented in an Italian historical tower of a non-conventional TMD [22].

In the present work, the results of a shaking table dynamic experiment conducted on a multi-degrees-of freedom frame structure equipped with a non-conventional TMD are presented. Compared to the literature experimental studies, in this paper the following innovative aspects are investigated. The effect of floor segmentation and isolation via a non-conventional TMD without adding structural mass on an original frame structure is primarily addressed; the effectiveness of the insertion of a TMD with a high mass ratio in an original structure is also investigated; a wide parametrical investigation on

structural dynamics as well as control capacity, varying the type and excitation intensity, is carried out; the application of a nonlinear non-conventional TMD as a robust and effective control system for structural mitigation is assessed. Preliminary results regarding structural dynamic characterization were reported in [23], whereas in this paper full results of the experimental campaign are presented. The control system is realized by isolating the structure top story mass of a four-story steel frame structure, converting it into tuned mass, and connecting it to the substructure with two high damping rubber bearings placed in series. Two objectives are to investigate aspects related to the dynamic structural response and to assess the seismic effectiveness of the non-conventional TMD for control purposes. Three structural configurations are in turn investigated concerning the dynamic response and control effectiveness: (1) the reference four-story structure, (2) the three-story intermediate structure, (3) the three-story structure equipped with the non-conventional TMD. The TMD design according to a well-established literature design procedure is preliminary illustrated. The input motion conditions, considered at different intensities, are: white noise, sine sweep, and natural earthquakes. The results of the experimentation at each input aim at demonstrating that a non-conventional TMD implemented via inter-story isolation is a smart control strategy useful for enhancing structural vibration mitigation. Moreover, since the adopted isolators, i.e., high damping rubber bearings, introduce nonlinearity in the system, the nonlinear effects provoked on the frame story structure are investigated throughout the paper.

The paper is organized as follows: Section 2 describes the physical models tested, Section 3 summarizes the non-conventional TMD design, Section 4 describes the experimentation, Section 5 illustrates the results of the experimentation based on the tests conducted (white noise, sine sweep, and natural earthquakes), and Section 6 reports the main conclusions of the work.

## 2. Description of the Physical Models

The original physical model is a 1:5 scale model steel frame structure ($\lambda_L = 5$). It consists of a four-story frame structure, with plan dimensions of 636 mm × 670 mm, an inter-story height of 600 mm, and made of L cross-section beams joined in order to form a T cross-section. The rectangular columns have dimensions of 80 mm × 8 mm, and the slabs are made by plates of dimensions 576 mm × 580 mm. The floor mass for each story is $m_i$= 131 kg ($i$ = 1, 4). Horizontal plates with two welded vertical orthogonal plates are bolted to the table realizing the base constraints of each rectangular column. The columns are bolted to the base constrains at one side only (Figure 2a). As a result, the columns are not symmetrically constrained. A picture of the physical model is shown in Figure 2a, whereas the model dimensions (plane and elevation) are reported in Figure 2b,c. Base motion is applied in the direction of minor inertia of the columns. The four-story frame structure represents the first configuration tested, named F4 (Table 1).

Configuration F4 is segmented into a three-story substructure and a one-story superstructure, which is converted into tuned mass (Table 1). A support plane, indicated with subscript "S", is built on the top of the three-story frame and connected by two cross bracings in the motion direction in order to realize a rigid slab to place the isolators. The three-story structure with the support plane, named F3S, represents the second intermediate configuration tested, and it is indicated in Table 1, where the detail of the cross bracings and the rigid support plane are evidenced. The non-conventional TMD is realized by placing the one-story mass, which represents the superstructure, on two high damping rubber bearings (HDRB) placed in series, positioned in the center of the support plane (Figure 3b). The three-story structure with a rigid support plane controlled by the non-conventional TMD implemented via inter-story isolation, named FTMD, is the third configuration tested (Table 1, depicted in Figure 3a). It has total height of 2.48 m, floor masses $m_1 = m_2 = 131$ kg and $m_3 = 133$ kg, a support plane mass $m_S = 17$ kg, and a TMD mass $m_{TMD} = 100$ kg. Model dimensions are reported in Figure 3c.

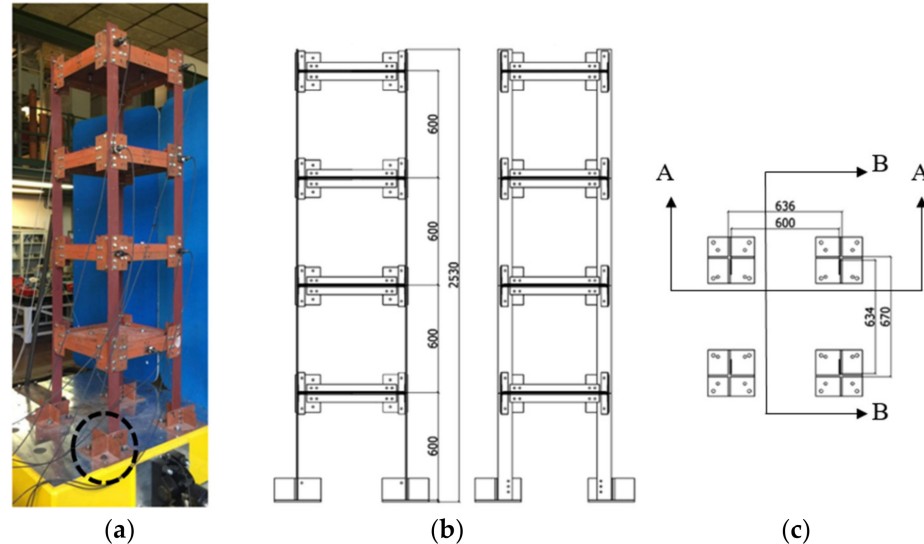

**Figure 2.** Four-story frame structure, configuration F4: (**a**) physical model, the circle indicates the detail of the columns' base constraint; (**b**) vertical sections, left: section A-A, right: section B-B; (**c**) plane section. Dimensions in millimeters.

**Table 1.** Configurations tested in the experimentation: F4—the reference 4-story structure, F3S—the 3-story structure with the rigid support plane; FTMD—the 3-story structure equipped with the non-conventional TMD.

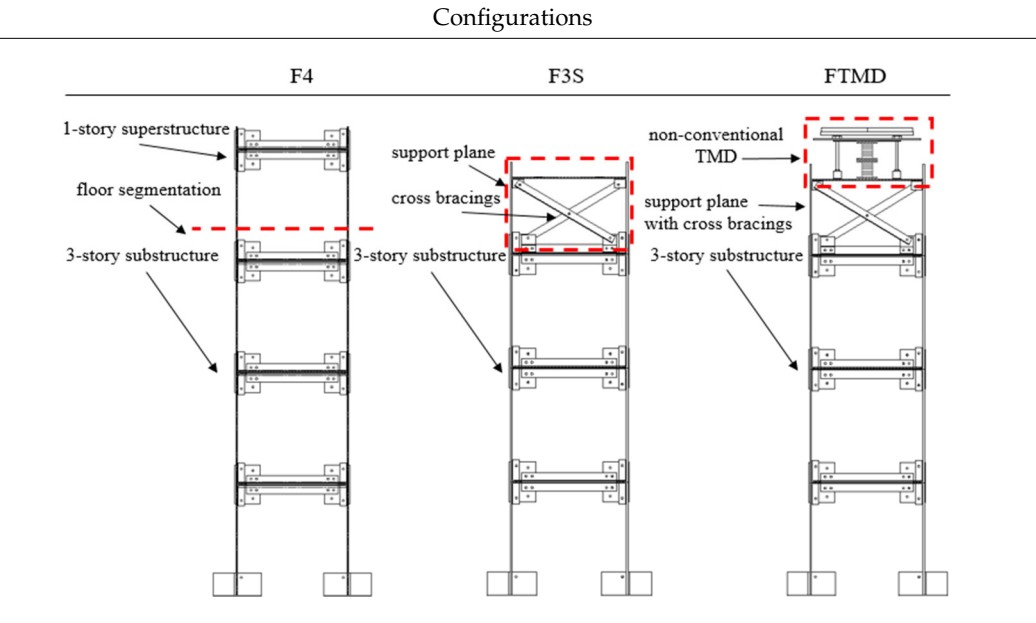

The HDRB isolators, realized according to the TMD design described in Section 3, consist of two bearings made by 27 layers, 2 mm thick, and 26 steel shims, 1 mm thick. The total rubber thickness is 54 mm, the height of the bearing is 80 mm, and the total diameter is 58 mm, including 53 mm of shim diameter and 5 mm of cover (Figure 4). A truss structure connected at the base of the support floor by four spherical bearings (see Figure 3b) is realized in order to avoid flexural behavior of the isolators, which sustain only shear strain. The threaded rods are braced in the direction of the motion.

The configurations tested are summarized in Table 1: (1) F4—the reference four-story structure, (2) F3S—the three-story structure with the rigid support plane, (3) FTMD—the three-story structure equipped with the non-conventional TMD. Concerning the investi-

gation of the dynamic structural response, the FTMD configuration is compared with the two reference cases: F4, the four-story structure, and F3S, the three-story structure with the support plane.

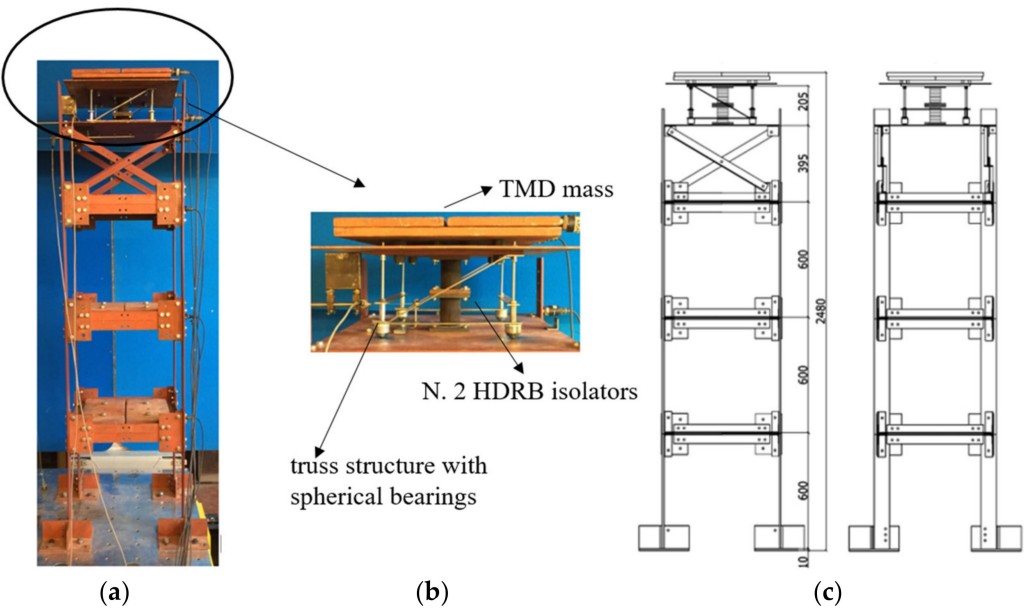

**Figure 3.** Three-story frame structure with support plane and non-conventional TMD, configuration FTMD: (**a**) physical model, (**b**) detail of the non-conventional TMD, (**c**) vertical sections, dimensions in millimeters.

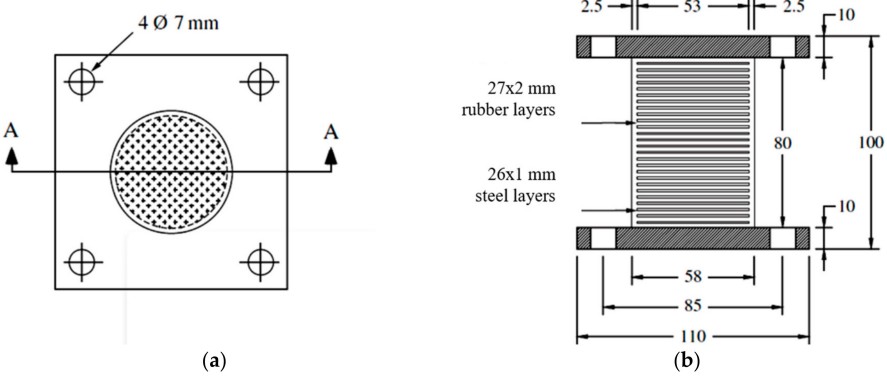

**Figure 4.** HDRB isolator: (**a**) plant view, (**b**) section A-A.

## 3. Non-Conventional TMD Design

The non-conventional TMD has been designed according to the well-established literature design procedure described in [3].

Given the four-degrees-of-freedom (DOF) frame structure, we model by taking the masses lumped at floor level and assuming purely translational motion. The isolation system located below the fourth floor segments the four-story frame structure into a lower (L) portion, also denominated as substructure, and an isolated upper (U) portion, also denominated as superstructure, corresponding to a number of DOF indicated as $N_L = 3$ and $N_U = 1$ DOF, respectively (see in Figure 5a). With reference to this physical model, it is possible to utilize a reduced-order 2-DOF model for design purposes (Figure 5b): the first DOF with mass $m_1$, stiffness $k_1$, and damping $c_1$, represents the generalized primary structure, F3S configuration (Table 1), and the second one represents the non-conventional TMD with mass $m_2$, stiffness $k_2$, and damping $c_2$.

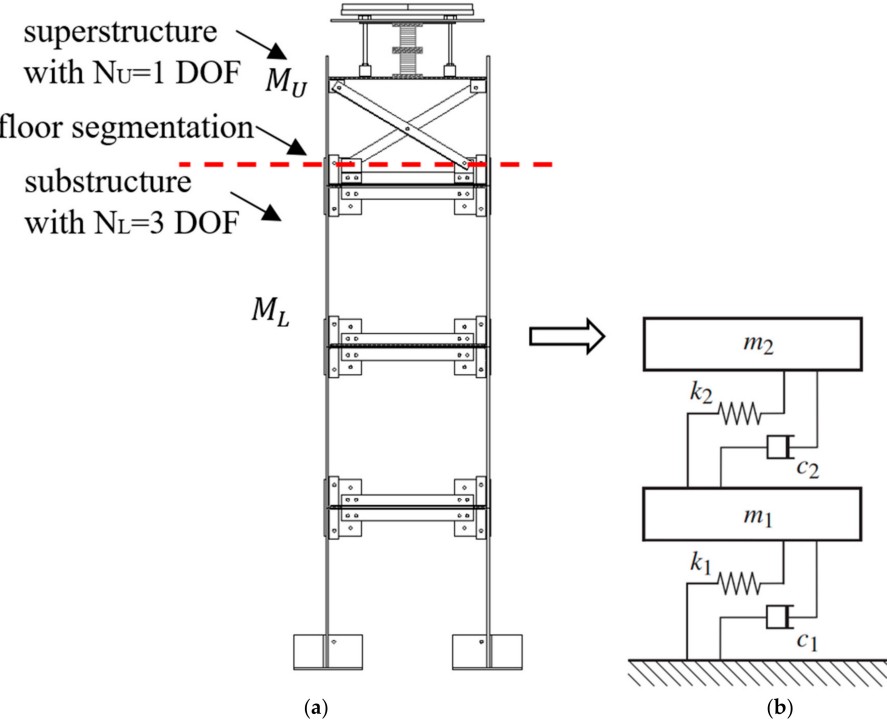

**Figure 5.** Structural models of the frame structure with inter-story isolation below the 4th floor: (**a**) 4-DOF model; (**b**) 2-DOF reduced-order model. Red dotted line indicates floor segmentation.

The response quantities and dynamic properties of the reduced 2-DOF model are evaluated as reported in [3]. The design parameters of the isolation system ($k_2$, $c_2$) that has a TMD mass $M_U = m_2$ = 100 kg were determined according to the optimization problem proposed in [3], where an energy performance index was defined and maximized in the space of the system parameters.

Based on the three-story physical model defined in Section 2, the dynamic properties of the generalized primary structure obtained through a finite element model are: the natural frequency and damping ratio, evaluated as $\omega_1 = \sqrt{k_1/m_1} = 25.06$ rad/s and $\zeta_1 = c_1/\left(2\sqrt{k_1 m_1}\right) = 0.01$, respectively, with $m_1 = 253$ kg being the first mode modal mass. For the TMD the following non-dimensional parameters were evaluated: the mass ratio $\mu = m_2/m_1$ and the frequency ratio $\alpha = \omega_2/\omega_1$, where $\omega_2 = \sqrt{k_2/m_2}$ represents the uncoupled frequency of the secondary oscillator and the secondary oscillator uncoupled damping ratio $\zeta_2 = c_2/\left(2\sqrt{k_2 m_2}\right)$. In the study, the mass ratio was valued as $\mu = 0.39$, whereas the obtained design parameters were $\alpha_{opt} = 0.7$ for a $\zeta_{2opt} \approx 0.10$, which correspond to the TMD stiffness and damping coefficient, respectively, of $k_2 \cong 30$ kN/m $c_2 \cong 0.35$ kNs/m, furnished by two HDRBs placed in series, as depicted in Figure 3b.

## 4. Experimentation

### 4.1. Experimental Set-Up

Tests were carried out on a one-degree-of-freedom 1.5 × 1.5 m shaking table L081-324-011 by MOOG Company (Elma, NY, USA) in the Laboratory of Sapienza University of Rome, the characteristics of which are reported in Figure 6a.

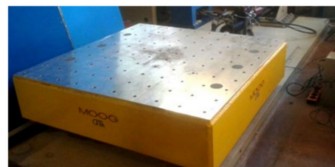

| Performance | |
|---|---|
| Frequency range | 0–25 Hz |
| Maximum stroke length | 0.40m |
| Servo-valve | 11 L/min–160 L/min |

(**a**)

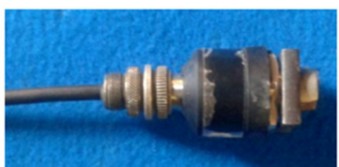

| Performance | |
|---|---|
| Sensitivity (±5%) | 1000 mV/g |
| Measurement range | ±5 g pk |
| Frequency range (±5%) | 0.5–2000 Hz |
| Broadband resolution | 0.00001 g rms |

(**b**)

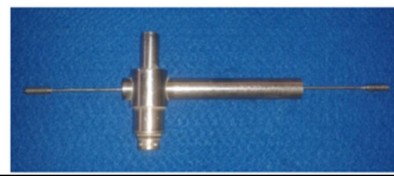

| Performance | |
|---|---|
| Sensitivity | 80 mV/V |
| Linearity deviation | ≤1% |
| Carrier frequency | 5 kHz |
| Nominal temperature | −200–100 °C |
| Ohmic resistance | 2 × 24 Ω |

(**c**)

**Figure 6.** (**a**) Shaking table, (**b**) piezotronic accelerometer, (**c**) Linear Variable Displacement Transducer.

The following instruments were utilized in order to determine the structural response: n. 8 PCB piezotronic accelerometers (Figure 6b) and n. 2 LVDTs—Linear Variable Displacement Transducers (Figure 6c). The instruments were positioned as indicated in Figure 7, for the exemplificative case of the FTMD configuration.

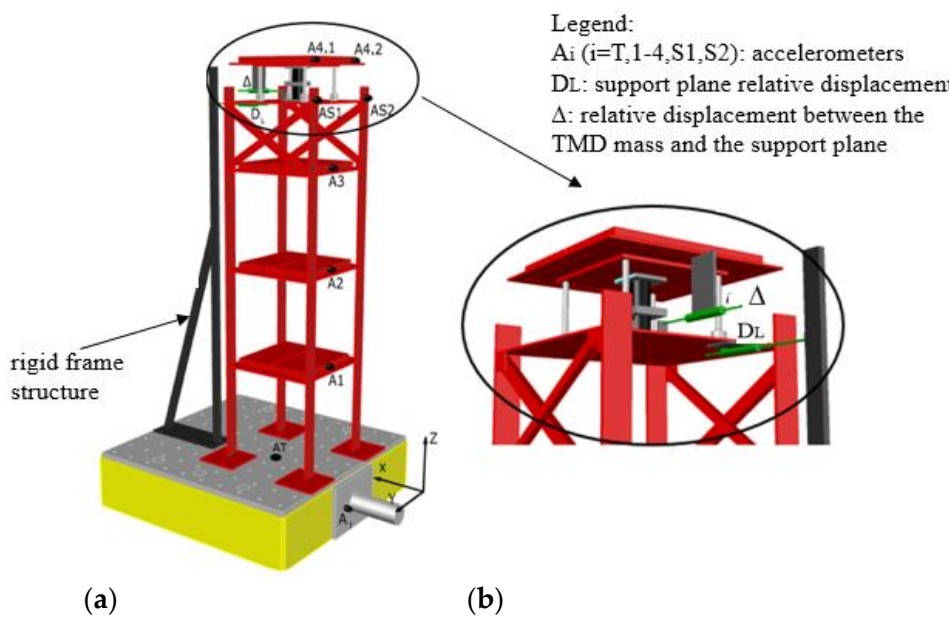

(**a**)　　　　　　(**b**)

**Figure 7.** Position of the measurement instruments on the FTMD configuration: (**a**) 3-story structure with non-conventional TMD, (**b**) detail of the TMD.

One accelerometer measured the shaking table (AT) and the others the structural accelerations, one at the first three levels (A1, A2, A3), and two on the support plane and on the TMD mass, respectively (AS1, AS2, A4.1, A4.2) (Figure 7a). One LVDT measured the relative displacement of the support plane with respect to a rigid frame located near the structure, named $D_L$, the other measured the inter-story drift between the TMD mass and the support plane, named $\Delta$ (Figure 7b). The TMD relative displacement with respect to the shaking table $X_{TMD}$ was therefore evaluated as $X_{TMD} = D_L + \Delta$.

For data acquisition of the whole system, the acquisition control unit MGCplus by HBM was utilized, with a sampling rate of 100 Hz. Each channel had filters to reduce noise and to guarantee high-quality data. The signals were finally processed in a Personal Computer.

### 4.2. Input Signals

Shaking table tests consisted of both dynamic identification and seismic tests. The input signals utilized were: (i) white noise tests (WN), (ii) increasing and decreasing sine sweep (SS) tests, (iii) natural earthquake (NE) records considering two near-fault and two far-field earthquakes. The description of each signal with the range of frequency and intensity investigated are reported in detail in Tables 2 and 3. For the earthquake signals, the time axes were scaled by the factor $\lambda_T = \sqrt{1/\lambda_L} = 0.45$.

**Table 2.** Input signals utilized in the experimentation.

| Description | Type of Signal |
|---|---|
| White noise (WN) tests. Range of frequencies investigated 1–25 Hz, range of acceleration investigated with RMS values 0.002–0.04 g. | 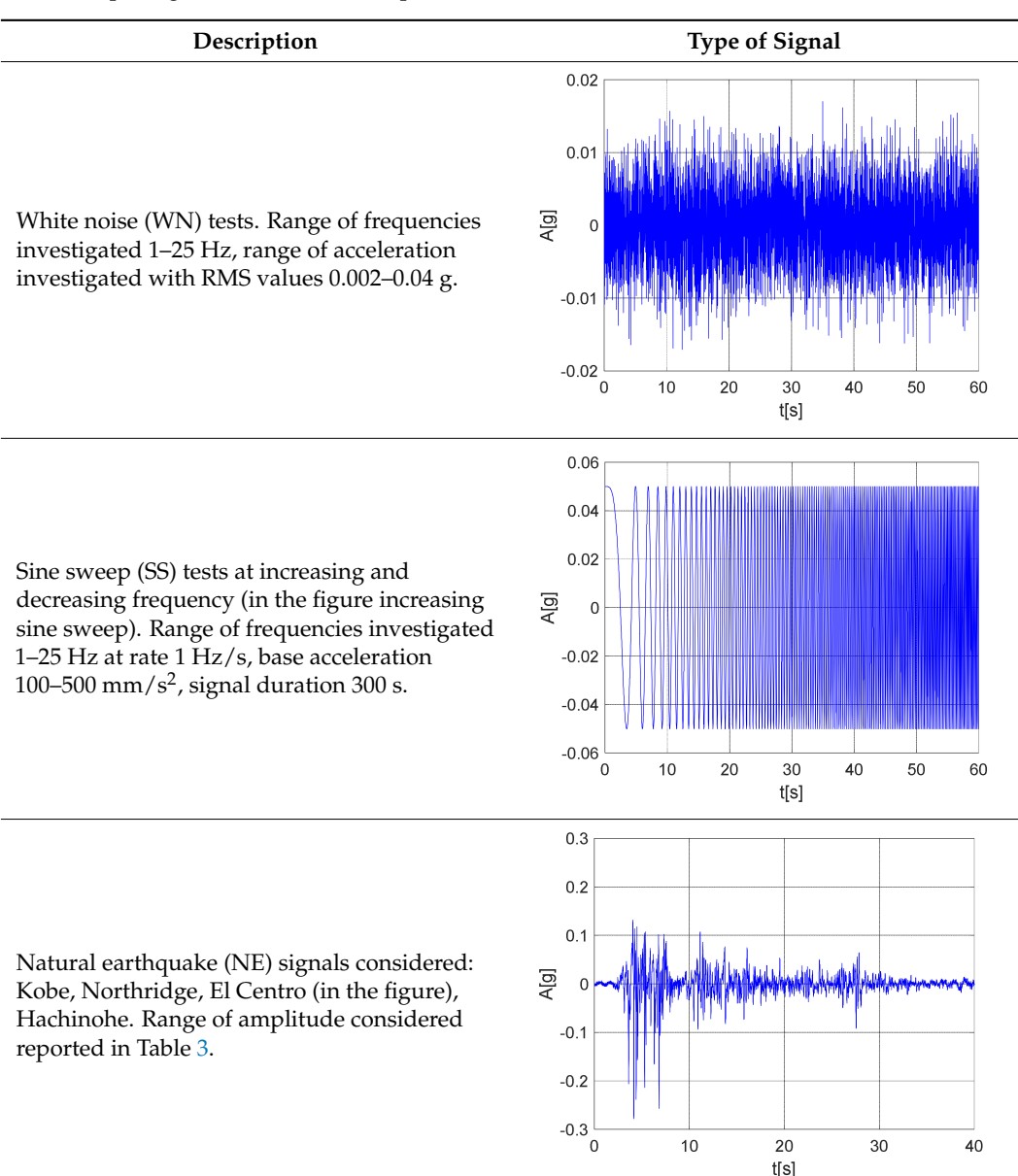 |
| Sine sweep (SS) tests at increasing and decreasing frequency (in the figure increasing sine sweep). Range of frequencies investigated 1–25 Hz at rate 1 Hz/s, base acceleration 100–500 mm/s$^2$, signal duration 300 s. | |
| Natural earthquake (NE) signals considered: Kobe, Northridge, El Centro (in the figure), Hachinohe. Range of amplitude considered reported in Table 3. | |

**Table 3.** Natural earthquakes' characteristics.

| Earthquake | PGA (g) | RMS (m/s$^2$) |
|---|---|---|
| Kobe | 0.05 | 0.07 |
| | 0.18 | 0.25 |
| | 0.23 | 0.32 |
| | 0.3 | 0.42 |
| Northridge | 0.06 | 0.05 |
| | 0.23 | 0.22 |
| | 0.28 | 0.27 |
| | 0.29 | 0.28 |
| El Centro | 0.05 | 0.06 |
| | 0.16 | 0.2 |
| | 0.26 | 0.3 |
| | 0.3 | 0.35 |
| Hachinohe | 0.05 | 0.07 |
| | 0.13 | 0.2 |
| | 0.16 | 0.26 |
| | 0.19 | 0.32 |
| | 0.25 | 0.37 |

## 5. Experimental Results

### 5.1. Dynamic Tests

The dynamic response and system properties of the tested structures were obtained through analysis of the experimental pseudo-frequency response functions, PFRFs, in the case of white noise input. The tests were repeated at different intensities to investigate possible emerging nonlinear effects due to the insertion of the non-conventional TMD via inter-story isolation with HDRBs.

The experimental PFRFs of the absolute acceleration and relative displacement at the support level are depicted in Figure 8a,b, respectively, for the F3S configuration at a given input intensity (F3S, black curve) and for the FTMD configuration at three increasing input intensities (green WN01 = 0.002 g, blue WN05 = 0.008 g, and red WN12 = 0.025 g curves, respectively). Focusing the attention first on F3S, the uncontrolled configuration, five main amplifications are observed in the acceleration response (Figure 8a), evidencing additional modes due to asymmetric base boundary conditions. In the displacement response (Figure 8b), instead, three peaks are visible. For both responses, the first mode always has highest amplification, the following being less amplified.

By observing in the figures the responses obtained with the FTMD configuration, focusing the attention on the curve at the lowest intensity it is possible to observe that in both responses the first mode splits into two smaller amplitude modes of frequencies smaller and greater than the first uncontrolled one, respectively, dramatically reducing the first amplification of the three-story frame structure (black curve).

Observing more in detail, for the acceleration response (Figure 8a), the first two modes have similar amplitudes, whereas for the displacement (Figure 8b), the first one is always greater than the second. The subsequent modes instead are modestly attenuated, without varying appreciably the frequencies with respect to the uncontrolled case. By increasing the input intensity, it is possible to notice that the first two amplifications move to the left, evidencing the softening behavior induced in the structure by the presence of the HDRB isolators; moreover, the first amplification attenuates, while the second one increases. This result is connected to the damping effect and energy exchange between the two modes that, by increasing the intensity, increase for the first mode and decrease for the second one. In any case, the control action induced by the TMD is always effective in a sufficient wide range of frequencies centered on the first uncontrolled one.

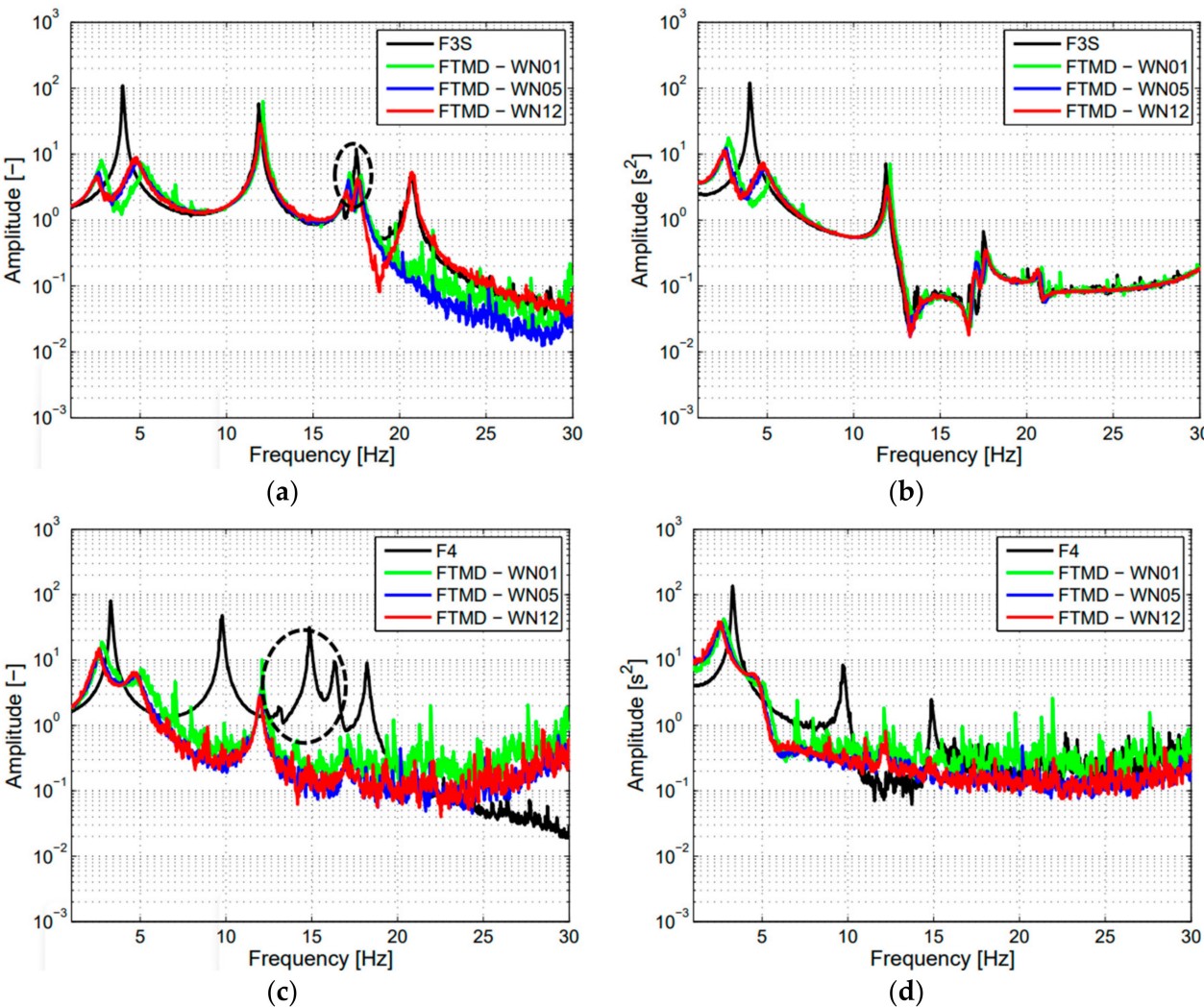

**Figure 8.** WN. Frequency response functions: (**a**) absolute acceleration at support plane, (**b**) relative displacement at support plane, (**c**) 4th floor absolute acceleration, (**d**) 4th floor relative displacement. Legend: F3S, F4, FTMD configurations at RMS amplitudes WN01 = 0.002 g, WN05 = 0.008 g, WN12 = 0.025 g. Circles in the figure indicate additional modes due to asymmetric base boundary conditions.

The experimental PFRFs of the fourth floor absolute acceleration and relative displacement are depicted in Figure 8c,d, respectively, for the F4 configuration (black curve) at a given reference intensity in comparison with the same responses obtained with the FTMD configuration at three input increasing intensities. In this comparison, the F4 configuration represents the non-segmented structure before the inter-story isolation has been implemented as control strategy. Concerning F4 configuration, in the acceleration response (Figure 8c), six amplifications are observed, evidencing additional modes due to asymmetric base boundary conditions. In the displacement response (Figure 8d), instead, only three peaks are visible. It is possible to observe that the first mode has the highest amplification, the following being less amplified, especially for the displacement response. By comparing the PFRFs of the non-segmented structure with the experimental PFRFs of the absolute acceleration and relative displacement at the fourth floor (TMD level) of the FTMD configuration, as a general result, it can be stated that the dynamics of the two systems greatly differ.

For the FTMD configuration both acceleration and displacement signals are strongly reduced with respect to the uncontrolled case: the first mode splits into two smaller

amplitude modes, with frequencies lower and higher than the first uncontrolled one, respectively, and amplifications dramatically reduced. The first peak generally is always greater than the second one, and this is more evident for the displacement response. A third modest amplification is evident only in the acceleration response (Figure 8c). All peaks subsequent to the first appear dramatically reduced with the insertion of the non-conventional TMD. By increasing the input intensity, the curves move to the left, evidencing the softening behavior induced by the HDRB isolators, already noticed in the responses at the support level; moreover, the first amplification attenuates, while the second one modestly increases. This phenomenon is due to the variations in the first two modal damping factors with the intensity, which produce an increase in damping in the first mode and a decrease in the second one. The control appears effective in all the range of frequencies of the uncontrolled non-segmented structure.

Figure 9 reports the force displacement cycles of the isolators in the FTMD configuration at the three input intensities investigated. It can be noticed that at low intensity the isolators behave as rigid plastic elements, the friction effects being predominant due to the interaction with the support plane. Instead, at higher intensities the cycles become elliptic, denoting that most of the damping is due to viscous effects. Moreover, it is possible to observe that the effective stiffness of the isolators decreases by increasing the input intensity. From the force displacement cycles, the secant stiffness of the isolators in the range of displacements considered ranges from 75 kN/m (Figure 9a) to 25 kN/m (Figure 9c).

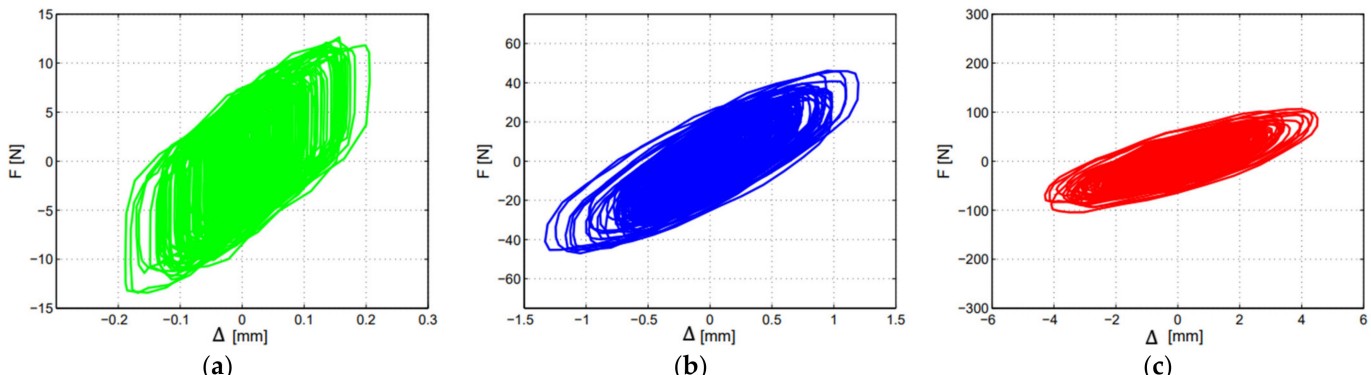

**Figure 9.** WN. FTMD configuration, force displacement cycles of the isolators: (**a**) WN01 = 0.002 g, (**b**) WN05 = 0.008 g, (**c**) WN12 = 0.025 g.

Making use of the experimental PFRFs shown in Figure 8, a procedure for the structural identification based on the ERA/OKID algorithm [24,25] was applied, providing an identified first-order representation of each configuration.

In the case of the F3S configuration, the identified relevant frequencies obtained are five, at about 3.98, 11.85, 16.71, 17.49, and 20.65 Hz, and the identified damping factors are around 0.48, 0.46, 0.40, 0.45, and 0.51 (%). While repeating tests at increasing intensities for this configuration, it was observed that frequencies modestly decreased, whereas damping factors had a modest increase; the results are not reported for sake of brevity. It can be concluded that the three-story frame structure with support was within the elastic range and behaved linearly.

In the case of the F4 configuration, the identified relevant frequencies are six, at about 3.25, 9.70, 12.95, 14.90, 16.30, and 18.25 Hz, and the identified damping factors are 0.92, 0.50, 0.40, 0.42, 0.44, and 0.34 (%). In addition, for the non-segmented structure, by increasing the input intensity, it was observed that the modes identified had well-spaced frequencies and modest decreases, whereas the damping factors had only a modest increase, evidencing that the four-story frame structure was within the elastic range and behaved linearly; the results are not reported for sake of brevity.

For the FTMD configuration, the dynamic properties at some different input intensities are evidenced in Table 4, where natural frequencies and damping factors for the six modes

identified are estimated. It is possible to notice that the first two frequencies decrease by increasing the intensity, whereas the subsequent ones have only a modest decrease. By observing damping factors instead, it can be noticed that the first one increases with the intensity while the second one decreases. In the subsequent modes, damping factors vary with the amplitude, but in minor way, always increasing, with the exception only of the third mode.

**Table 4.** FTMD—identified frequencies and damping factors varying input intensity.

| | Frequencies (Hz) Mode | | | | | |
|---|---|---|---|---|---|---|
| RMS (g) | 1 | 2 | 3 | 4 | 5 | 6 |
| 0.002 | 2.995 | 6.053 | 12.173 | 17.159 | 17.772 | 20.927 |
| 0.005 | 2.766 | 5.182 | 12.051 | 17.122 | 17.727 | 20.889 |
| 0.010 | 2.618 | 4.822 | 11.966 | 17.018 | 17.627 | 20.782 |
| 0.020 | 2.446 | 4.592 | 11.905 | 16.931 | 17.549 | 20.696 |
| 0.040 | 2.168 | 4.380 | 11.818 | 16.768 | 17.450 | 20.579 |
| | Damping Factors (%) Mode | | | | | |
| RMS [g] | 1 | 2 | 3 | 4 | 5 | 6 |
| 0.002 | 4.814 | 12.406 | 1.038 | 0.516 | 0.474 | 0.413 |
| 0.005 | 7.420 | 10.790 | 0.830 | 0.590 | 0.540 | 0.420 |
| 0.010 | 10.991 | 9.105 | 0.775 | 0.653 | 0.556 | 0.457 |
| 0.020 | 12.487 | 7.187 | 0.786 | 0.802 | 0.607 | 0.560 |
| 0.040 | 13.420 | 5.580 | 0.850 | 1.060 | 0.670 | 0.680 |

Figure 10a,b show, for the FTMD configuration, the variation in the first two frequencies, $f_1$ and $f_2$, and damping factors, $\eta_1$ and $\eta_2$, respectively, versus the RMS value of the base acceleration in the whole range investigated. In each sub-figure, the discrete values represent those experimentally identified, and the continuous curves instead represent their interpolations. By observing Figure 10a, the softening behavior introduced by the nonlinear TMD emphasized by the decrease in the first two frequencies is evident, which maintains a difference between them that is almost constant by increasing the excitation. The two curves have a regular trend and decrease with the intensity, maintaining almost the same distance between them. The first and second frequency tend to the limit values of 2.2 and 4.4 Hz, respectively, for the maximum intensity applied. The first frequency of the compared uncontrolled cases (dotted lines f1-F4 and f1-F3S in Figure 10a) are always contained in the variation frequency range of the two frequencies in the controlled case.

Considering the damping factors (Figure 10b), it is possible to observe the high dissipative capabilities induced by the HDRBs. The first damping factor increases with the intensity, reaching the limit value of 13.4%, whereas the second one decreases and tends to the limit value of 5.6%. A rapid exchange of damping between the two modes, which moves from the second to the first one, is observed by increasing the excitation. The energy transfer is evident between the two controlled modes, even if the mean value between them estimated at the various intensities maintains almost the same for both modes (in mean varying the intensity indicated by $\eta_{mean}$ Figure 10b $\approx$ 9–10%). The importance of appropriately setting modal damping by a TMD system in order to achieve its efficiency is analytically treated in [26].

In addition to white noise tests, increasing and decreasing sine sweep tests were conducted for the dynamic characterization of the three-story structure equipped with a non-conventional TMD. A typical time history registered by one accelerometer placed along the structure for increasing and decreasing SS test is depicted in Figure 11. When the SS passes through the resonance frequencies of the structure, a rapid increase in the response is observed. By taking the maxima and knowing the variation in the excitation frequency, it is possible to estimate the structural resonances directly from the time histories. Specifically, in the figure, it is possible to notice that the time history applied increasing in the first

150 s and decreasing in the last 150 s is not perfectly symmetrical due to nonlinearities introduced by the HDRB isolators. The corresponding resonances observed with increasing and decreasing sine sweep are indicated in Figure 11 with the same symbols. It is possible to observe that the amplifications are slightly different.

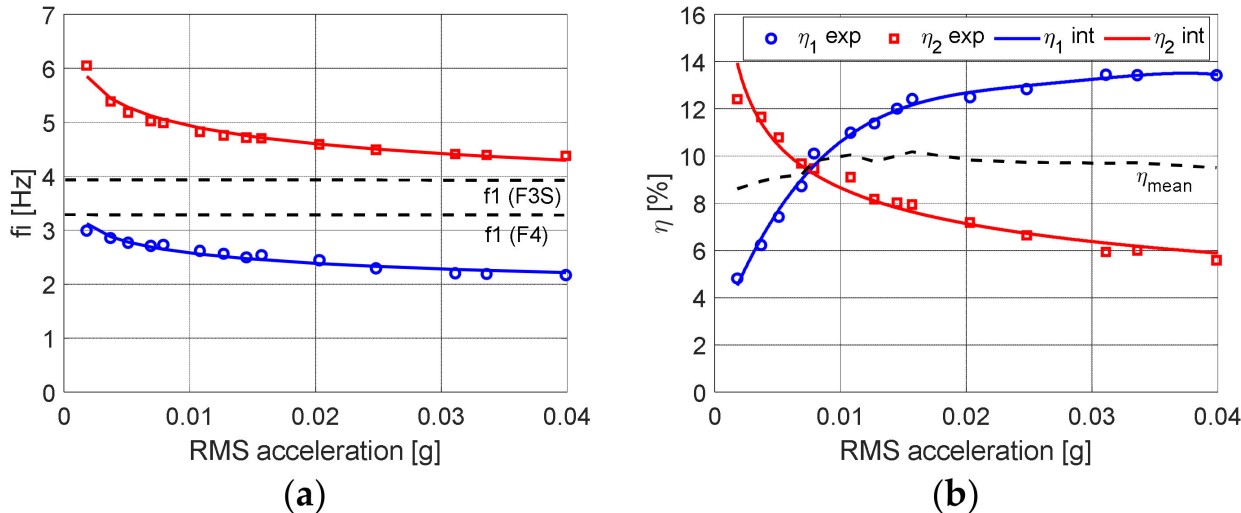

**Figure 10.** (**a**) First two identified frequencies for FTMD configuration versus the RMS of the base excitation (experimental values and interpolating curves), dotted lines: first frequency of F3S and F4 configuration, respectively, (**b**) first two identified damping factors for FTMD configuration versus the RMS of the base excitation (experimental values and interpolating curves), dotted line: $\bar{\eta}$ mean value of the first two identified damping factors versus the RMS of the base excitation.

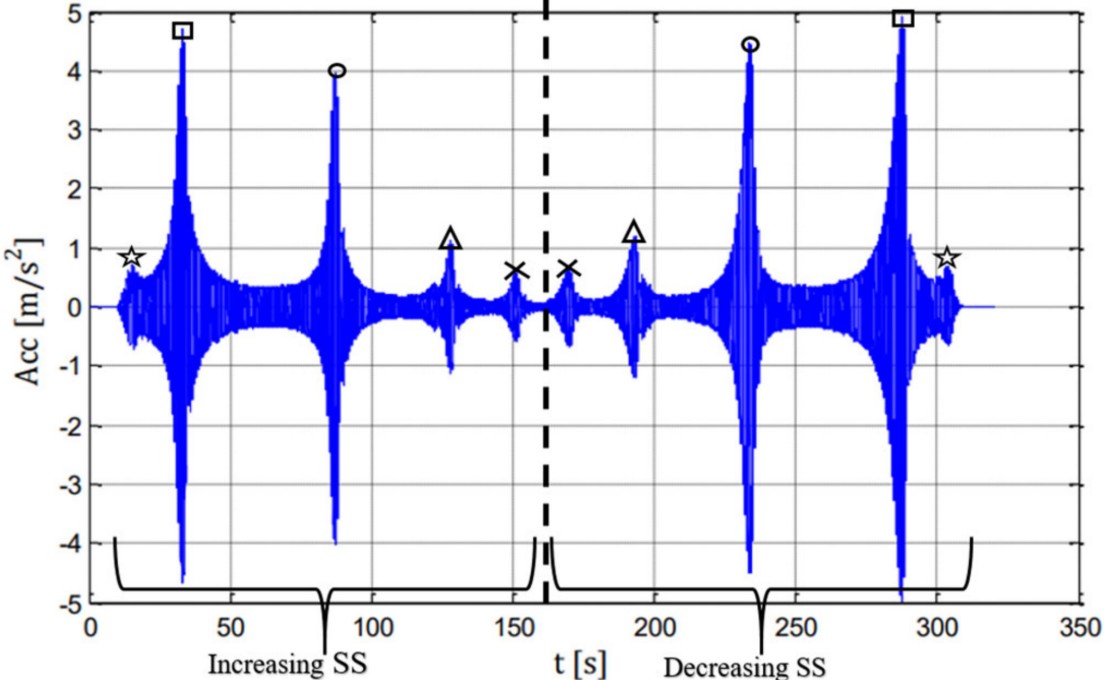

**Figure 11.** SS. FTMD configuration absolute acceleration at first floor for increasing and decreasing sine sweep. Stars, squares, circles, triangles, × represent same resonances at increasing and decreasing sweep respectively.

Figure 12a–d show the time histories with increasing SS of absolute acceleration at the support level, relative displacement at the support level, absolute acceleration at the fourth floor, and relative displacement at the fourth floor, respectively. Six resonance frequencies are visible from the support level absolute acceleration time history (Figure 12a), as was observed with the PFRF in Figure 8a. Three resonance frequencies are visible from the relative displacement at the support level (Figure 12b), as observed with the PFRF in Figure 8b, and the absolute acceleration at the fourth floor (Figure 12c), as indicated with the PFRF in Figure 8c. However, only the first two frequencies are visible from the relative displacement at the fourth floor (Figure 12d), as observed with the PFRF in Figure 8d.

Figure 13 reports details of the time history of the inter-story drift between the TMD mass and the support plane overlapped to the support level relative displacement time history, in the range where the first two resonance frequencies of the structure in the FTMD configuration appear. In the figure, the first two frequencies of the FTMD configuration and the first frequency of the F3S configuration are reported as well.

It can be observed that when the structure vibrates with the first resonance frequency ($f_1$-FTMD = 2.25 Hz), the inter-story drift of the TMD (TMD, blue curve in Figure 13) is higher and in phase with respect to the relative displacement of the substructure (the support level displacement, red curve in Figure 13). Differently, when the structure vibrates with the second resonance frequency ($f_2$-FTMD = 4.25 Hz), the TMD drift is lower and out of phase with respect to the relative displacement of the substructure.

By representing more in detail the signals around the three frequencies indicated in Figure 13, it is possible to have an idea of the first two modes of vibration of the FTMD configuration examined in Figure 14a–c. Specifically, in the top of Figure 14a, the first mode of vibration of the FTMD configuration is reported, evidencing the TMD and support modal displacement: in this mode, the structure and the TMD are in phase, the TMD displacement is greater than the structure displacement which is highly damped (Figure 14a, center), and the force displacement cycle of the TMD is large and elliptic (Figure 14a, bottom). At the first resonance frequency of the uncontrolled three-story structure, it is possible to notice that the mode of vibration of the structure with the TMD (Figure 14b, top) is out of phase with the TMD and support displacements similar but with opposite signs (Figure 14b, center), and the force displacement cycle of the TMD is small and elliptic (Figure 14b, bottom). At the second resonance frequency of the uncontrolled three-story structure, the mode of vibration of the structure with the TMD (Figure 14c, top) is out of phase, with support displacement less damped in the second mode (Figure 14c, center), but the force displacement cycle of the TMD is large and elliptic (Figure 14c, bottom).

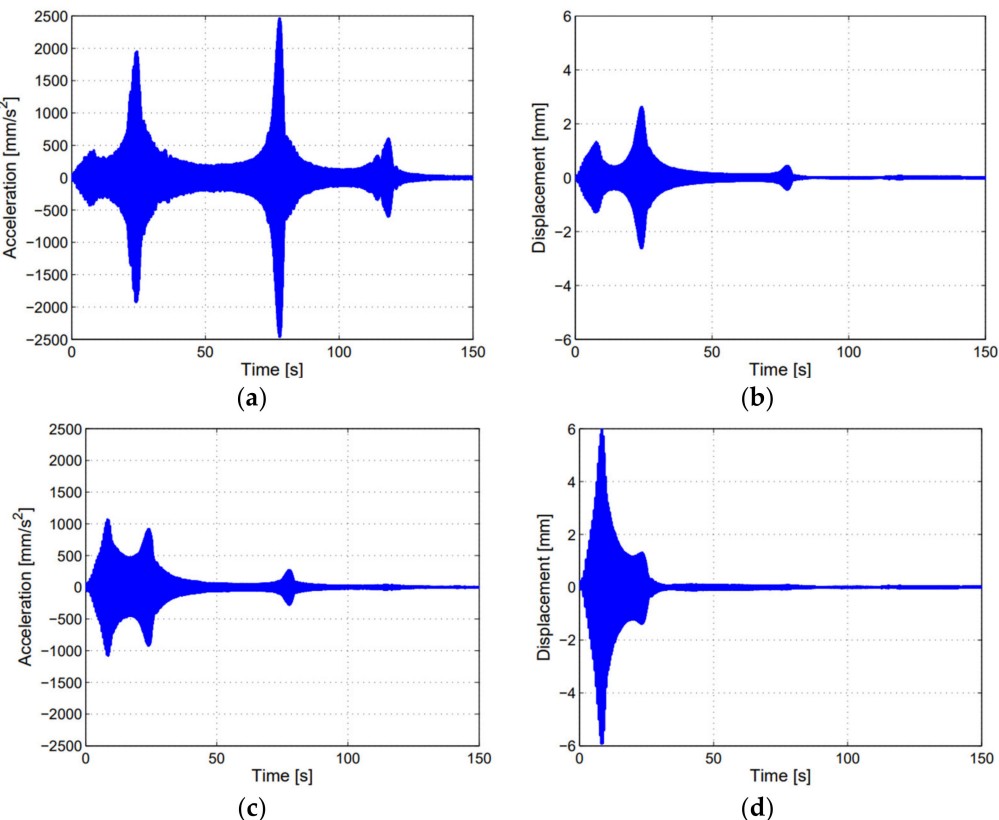

**Figure 12.** Increasing SS. FTMD configuration, time histories: (**a**) absolute acceleration at support level; (**b**) relative displacement at support level; (**c**) absolute acceleration at 4th floor; (**d**) relative displacement at 4th floor.

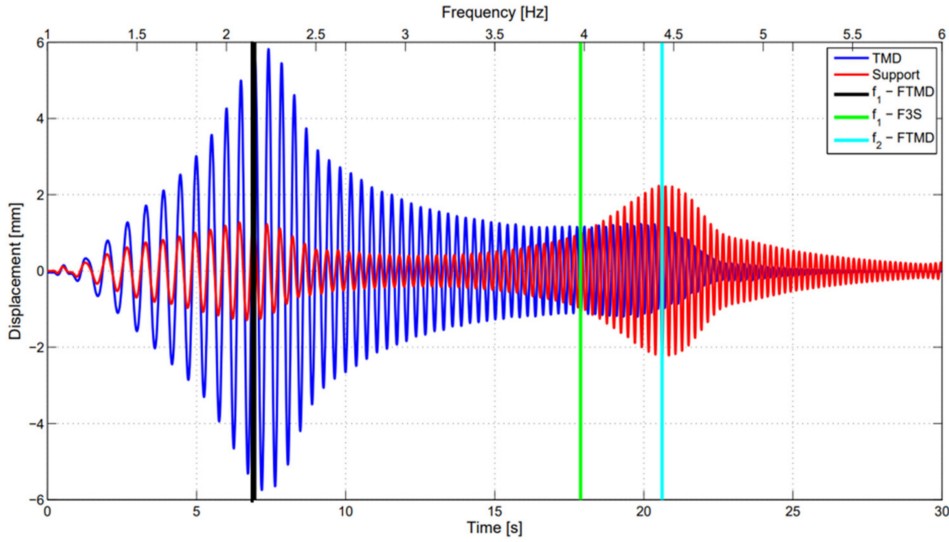

**Figure 13.** SS. FTMD configuration, detail of the time history of the inter-story drift of the TMD (blue curve) and the relative displacement at support level (red curve).

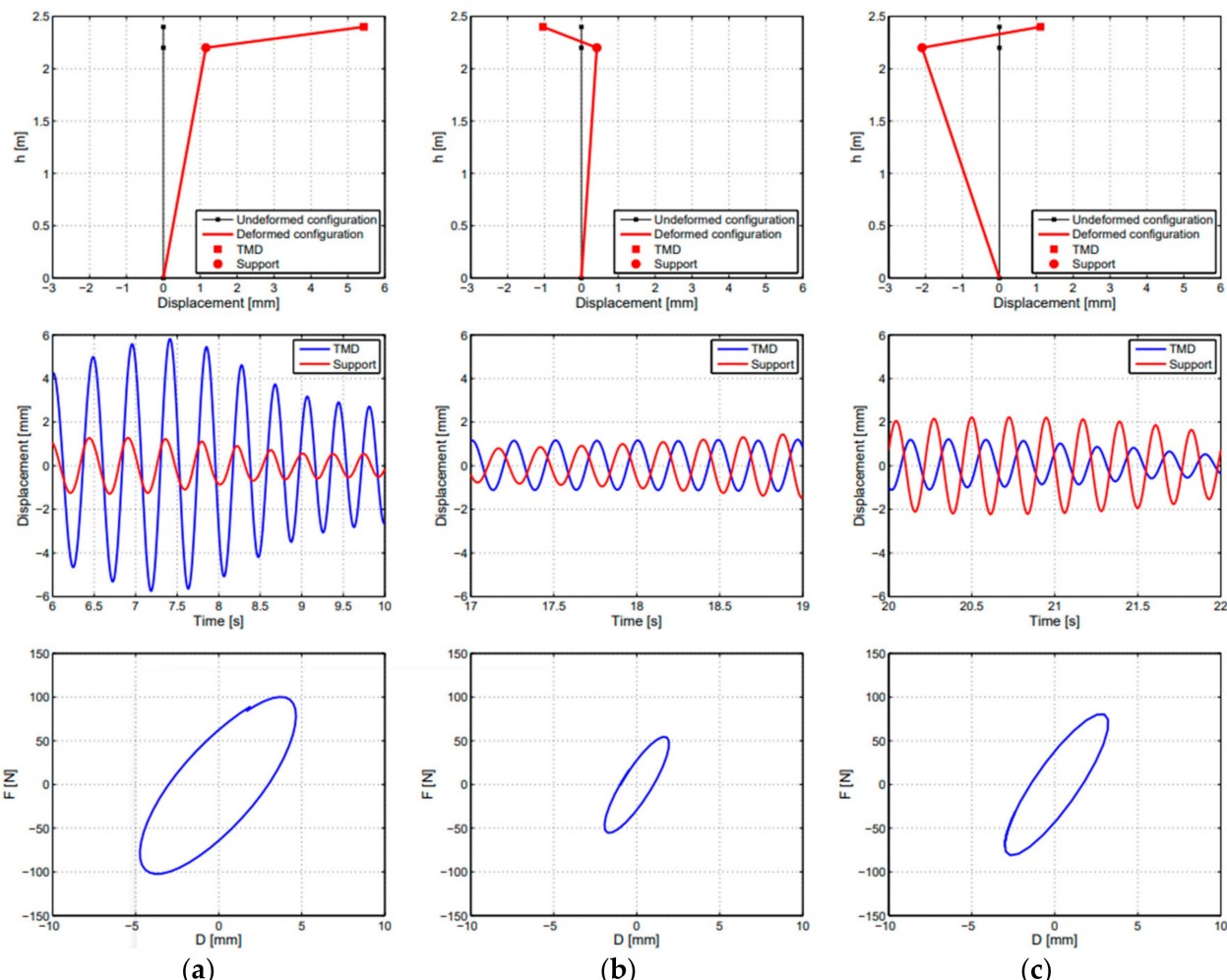

**Figure 14.** SS. FTMD configuration, detail in the increasing frequency range 1–6 Hz: (**a**) first mode of the FTMD configuration, (**b**) first mode of the F3S configuration, (**c**) second mode of the FTMD configuration. Top: modal displacements vs. structure height; center: TMD and support displacement time history; bottom: force displacement cycles of the HDRB isolators.

*5.2. Seismic Effectiveness*

The seismic effectiveness of the structure equipped with a non-conventional TMD was with the four natural earthquake signals of characteristics reported in Table 3, applied at increasing intensity. The three configurations were tested. A first comparison was made between the responses obtained with the FTMD configuration and the F4 configuration.

In order to represent synthetically the performance obtained with the use of the TMD in comparison with the non-segmented structure, the following response indices, $J_n$, $n = 1$–6, evaluated from the maximum values of each response quantity in the FTMD configuration normalized with the respect the same quantity in the F4 configuration, were defined:

$$
J_i = \frac{\max[abs(A_{i,FTMD})]}{\max[abs(A_{i,F4})]} \; i = 1,2,3, \; J_4 = \frac{\max[abs(D_{3,FTMD})]}{\max[abs(D_{3,F4})]},
$$
$$
J_5 = \frac{\max[abs(T_{b,FTMD})]}{\max[abs(T_{b,F4})]}, \; J_6 = \frac{\max[abs(M_{b,FTMD})]}{\max[abs(M_{b,F4})]}
$$

(1)

where $A_i$ represents the acceleration at the *i*-level, $D_3$ is the relative displacement at the third level, $T_b$ is the base shear, and $M_b$ is the base moment.

In the same manner, the response indices $I_n$, $n = 1$–$6$, evaluated from the RMS values of each response quantity in the FTMD configuration normalized with the respect the same quantity in the F4 configuration, were defined:

$$I_i = \frac{\text{RMS}(A_{i,FTMD})}{\text{RMS}(A_{i,F4})} \; i = 1, 2, 3, \; I_4 = \frac{\text{RMS}(D_{3,FTMD})}{\text{RMS}(D_{3,F4})},$$

$$I_5 = \frac{\text{RMS}(T_{b,FTMD})}{\text{RMS}(T_{b,F4})}, \; I_6 = \frac{\text{RMS}(M_{b,FTMD})}{\text{RMS}(M_{b,F4})} \tag{2}$$

Note that indices that have values lower than the unity imply the effectiveness of the control system in reducing the structural response in terms of RMS and peak values.

The response indices evaluated at different intensities are reported in Figures 15 and 16 for near-field and far-field earthquakes, respectively.

Considering the Kobe earthquake, in Figure 15a it can be shown that the indices evaluated in terms of peak values at first increase and then decrease with the PGA. Since the F4 configuration behaves linearly, these results highlight the nonlinear behavior of the structure controlled with a non-conventional TMD. All the indices have values lower than the unity, demonstrating the effectiveness of the TMD in reducing the structural responses. The reduction observed from the floor accelerations through indices $J_1$, $J_2$, and $J_3$ increases with the height, with percentages of reductions in the range of 60–75%. Concerning the displacement reduction, index $J_4$ modestly varies with the PGA, showing a reduction around 75%. Base shear and base moment, $J_5$ and $J_6$, have the same trend versus the PGA and show reductions of 75–80%. The indices evaluated in terms of RMS values modestly increase with the PGA and generally have higher reductions compared to the corresponding ones evaluated in terms of peak values.

The accelerations, indices $I_1$, $I_2$, and $I_3$, have percentages of reductions of 75–85%, whereas displacement, base shear, and base moment, $I_4$, $I_5$, and $I_6$, have reductions from 85% to almost 90%. In Figure 15c,d, for the Northridge earthquake, a similar trend of the response indices in terms of peaks and RMS values versus the PGA is observed. However, considering the acceleration peaks, $J_1$, $J_2$, and $J_3$, the reductions observed are generally lower, with percentages of reductions in the range of 25–75%. Instead, the displacement and the base shear and base moment show more reductions compared to accelerations that decrease with the PGA. The same responses evaluated in terms of RMS (Figure 15d) show a percentage of reduction that is almost constant with the PGA, especially for displacement, base shear, and base moment, which have reductions around 80%. Instead, accelerations are mostly reduced at higher floors, with reductions that decrease with the PGA.

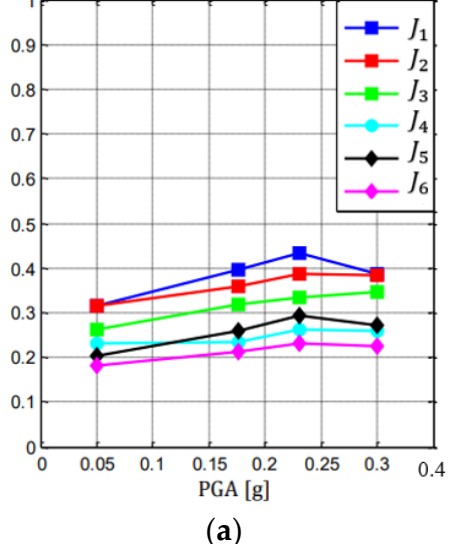

(a)

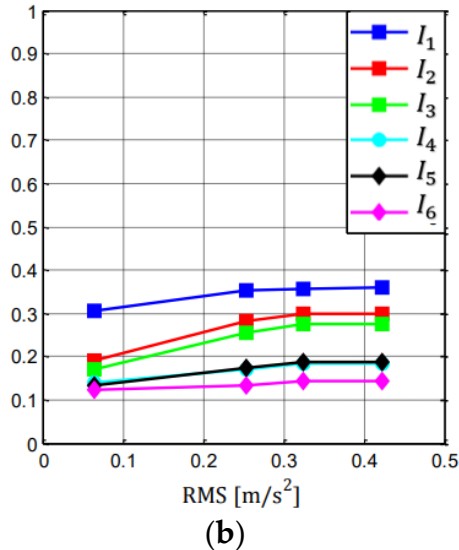

(b)

**Figure 15.** *Cont*.

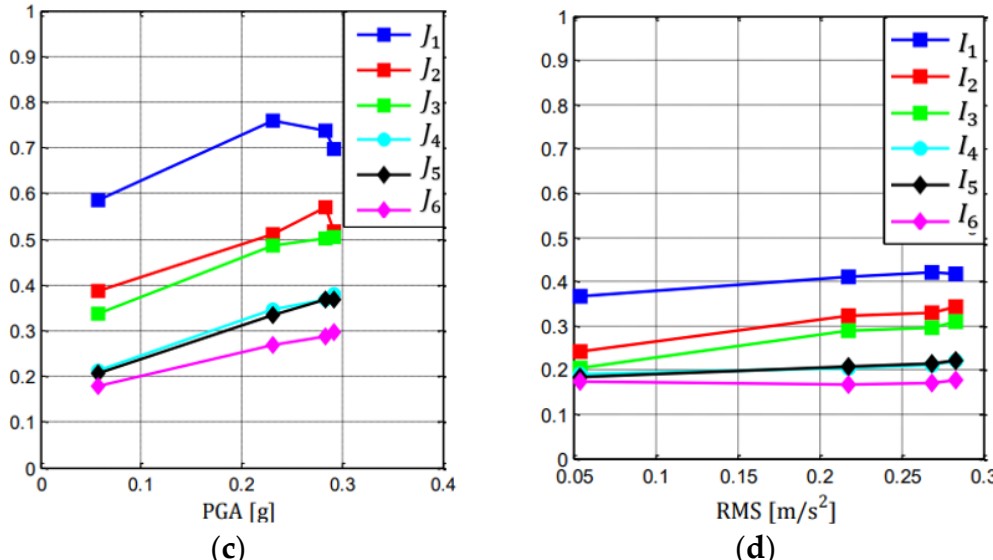

**Figure 15.** NE—response indices based on maximum values ($J_n$) and RMS values ($I_n$), $n = 1$–$6$, varying the input intensity for near-fault earthquakes: (**a**,**b**) Kobe, (**c**,**d**) Northridge.

Considering the far-field earthquakes, in Figure 16a,b the response indices in terms of peak and RMS values are depicted for El Centro. The acceleration peak values, $J_1$, $J_2$, and $J_3$, do not have the same trend with the PGA varying the floor: the reductions oscillate from a minimum of 50% to a maximum of almost 75%. Instead, displacement, base shear, and base moment have modest variations with the PGA, oscillating with reductions around the 70–75%. By looking at indices in terms of RMS (Figure 16b), it can be seen that $I_1$, $I_4$, $I_5$, and $I_6$ are almost constant with the PGA, with reductions of around 60% for the first floor acceleration and around 80% for the displacement, base shear, and base moment. Instead, the second and third floor acceleration , $I_2$, and $I_3$, have almost the same reductions, decreasing with the PGA from almost 85% to 65%. Figure 16c,d depict the response indices in terms of pack and RMS values for the Hachinohe earthquake. Generally, considering the peak indices (Figure 16c), as observed for El Centro, for the accelerations $J_1$, $J_2$, and $J_3$, the trend of reduction is not regular with the PGA and the floor considered, with reductions from 40–55%. Considering the displacement, $J_4$ modestly varies with the PGA, with reduction from 55% to almost 70%. The base shear, $J_5$, is constant, with a reduction of almost 70%, whereas the base moment, $J_6$, slowly decreases with the PGA, with reductions around 70–75%. The same indices considering the RMS values (Figure 16d) instead have an almost constant trend with the PGA, with reductions of 55–70% for floor accelerations, $I_1$, $I_2$, and $I_3$, and around 80% for displacement, base shear, and base moment ($I_4$, $I_5$, and $I_6$).

*5.3. Further Considerations on the TMD Effectiveness*

Some further comments concerning the TMD seismic effectiveness can also be deduced from the observation of the PFRFs obtained with WN input, reported in Figure 8. With respect to a non-segmented structure, F4 configuration, the implementation of inter-story isolation and realization of a non-conventional TMD produced a great attenuation of the dynamic structural response at all the frequencies (see Figure 8c,d), proving that it is a smart control strategy useful for enhancing structural vibration mitigation. The enhancement is especially evident for the acceleration response, which possesses high amplifications not only around the first modes, as is typical for relative displacement (Figure 8d), but also at higher ones. All the amplifications are effectively reduced by the non-conventional TMD implementation (Figure 8c).

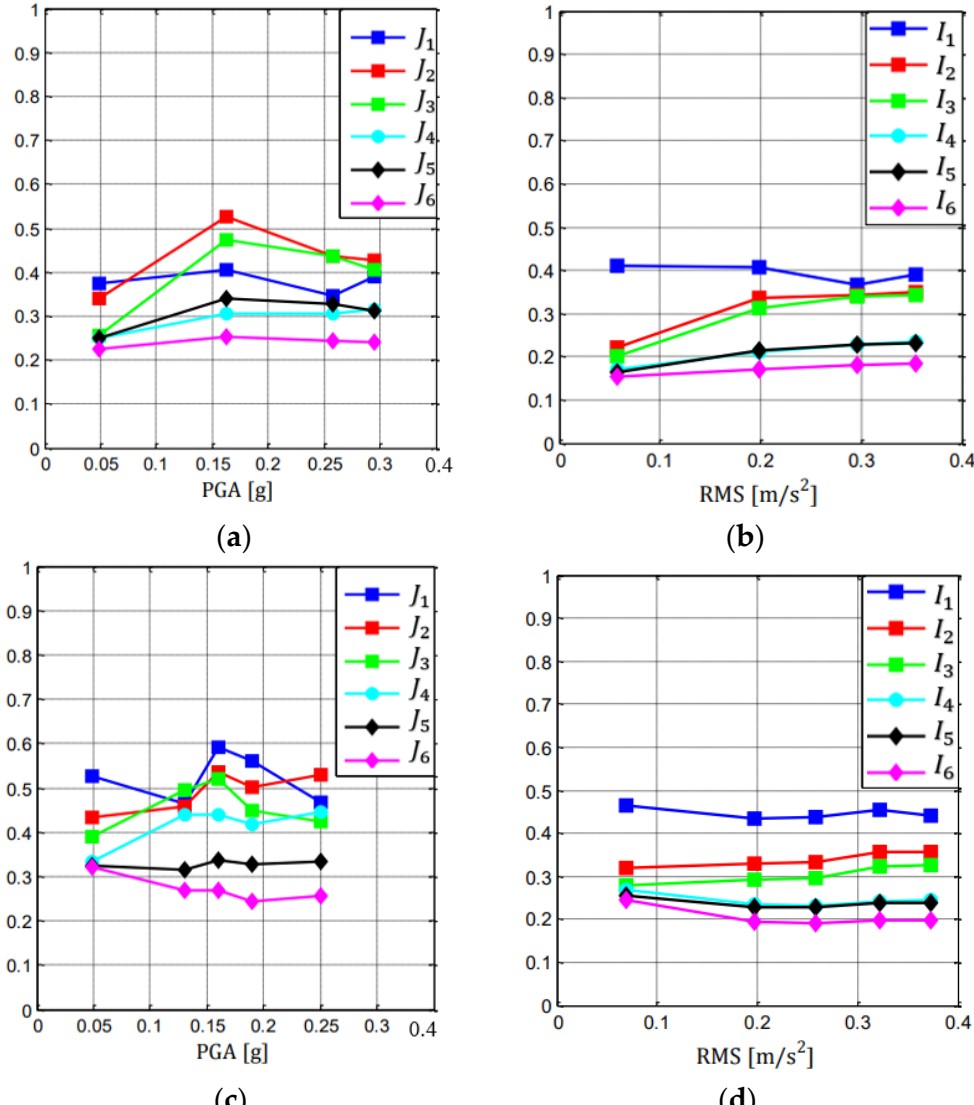

**Figure 16.** NE–response indices based on maximum values ($J_n$) and RMS values ($I_n$), $n$ = 1–6, varying the input intensity for far-field earthquakes: (**a**,**b**) El Centro; (**c**,**d**) Hachinohe.

The F3S configuration, instead, in this comparison can be viewed as a three-story primary structure without control. With respect to the F3S configuration, the introduction of a TMD with a large mass ratio (FTMD configuration) produced its control action mainly around the first mode, with a great attenuation of the dynamic response in a wide range of frequencies centered on the first uncontrolled mode. Subsequent modes instead were only modestly influenced by the TMD (Figure 8a,b).

The preliminary indications obtained with WN input were confirmed when the configurations were tested with multi-frequency actions as the earthquake signals.

The results are synthetically reported for two exemplificative response quantities, the maximum values of the third floor displacement $D_3$ and the base shear $T_b$ in Figure 17a,b, respectively, in the case of Kobe earthquake at different intensities, comparing the FTMD configuration with F4 and F3S configurations. In the case of F4 and F3S configurations, the responses have a linear trend, whereas in the case of the FTMD configuration the responses are modestly nonlinear. It is possible to notice that the insertion of the TMD is effective if applied to both cases for each response. In addition, the maximum effectiveness is obtained by implementing inter-story isolation from an original non-segmented structure (FTMD vs. F4 configuration).

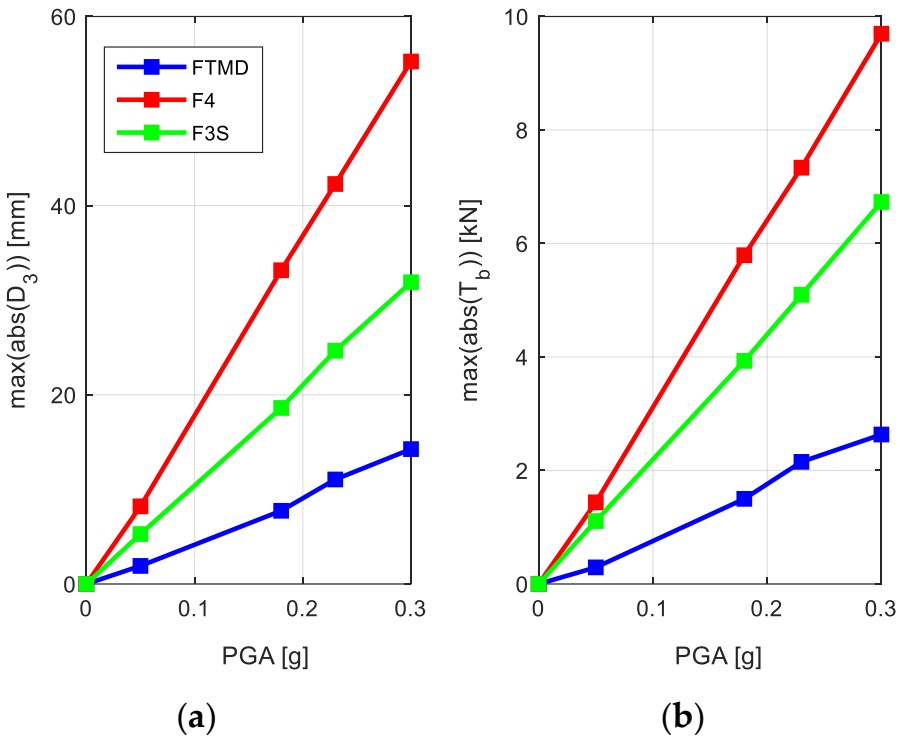

**Figure 17.** NE— Kobe earthquake, configurations compared: FTMD, F4, F3S, (**a**) maximum value of the third floor displacement $D_3$ versus PGA (**b**) maximum value of the base shear $T_b$ versus PGA.

In Figure 18a,b, as an exemplificative case, the time history of the base shear is reported in the case of the Kobe earthquake at a PGA level of 0.18 g, comparing FTMD with the F4 configuration, and FTMD with the F3S configuration, respectively. The base shear is evidently reduced with respect to both configurations considered. The response time history is attenuated from the very first pulses and has almost null oscillations after the first decades of seconds. In addition, observing the time history, the maximum effectiveness of the TMD applied to an original non-segmented structure with inter-story isolation is confirmed (Figure 18a).

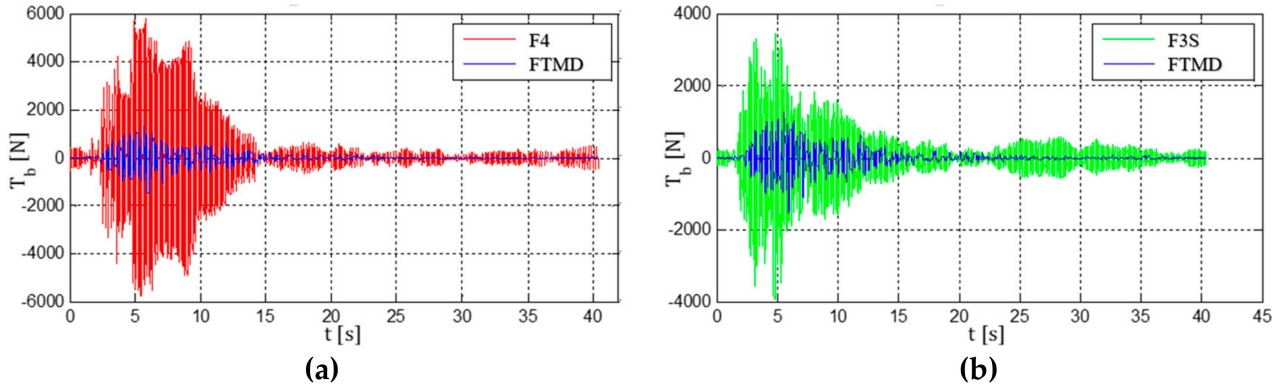

**Figure 18.** NE—time history of the base shear for Kobe earthquake at given intensity level, PGA = 0.18 g: (**a**) FTMD vs. F4 configuration, (**b**) FTMD vs. F3S configuration.

## 6. Conclusions

The results of a shaking table dynamic experiment conducted on a four-story frame structure equipped with a non-conventional TMD were presented. The control system was realized by isolating the top story mass of the frame structure and connecting it to the substructure with two HDRBs. Through two reference models for comparisons, tests

highlighted the effect of floor segmentation and isolation via a non-conventional TMD, without adding structural mass, on an original four-story frame structure, as well as the effect of the insertion of a non-conventional TMD as a general control application on a three-story structure. The main conclusions drawn are the following:

- From white noise tests, the dynamical characterization of the models tested was conducted by observation of PFRFs. For each configuration, structural identification was carried out. A dynamic behavior for the F4 and F3S configurations emerged that was almost linear. For the FTMD configuration, a softening behavior due to the nonlinearity introduced by the HDRB isolators emerged by increasing the excitation. The first two frequencies decreased with the input intensity, however they maintained a difference between each other, that was almost constant by increasing the excitation. Instead, the first two damping factors showed a rapid exchange between them that was almost constant in the mean value. The high dissipative capabilities induced by the HDRB isolators were highlighted.
- Sine sweep tests confirmed the main resonances observed for the controlled structure as well as the response nonlinearities, highlighted through different amplification values at increasing and decreasing tests.
- The effectiveness of the control strategy was proven from seismic tests, in addition to what was evidenced by white noise and sine sweep tests. Implementing inter-story isolation to realize a non-conventional TMD, all responses were strongly reduced in terms of peaks and RMS values in a wide range of intensities. Adding a non-conventional TMD in an original uncontrolled structure was demonstrated to be effective as well, producing large response reductions in all the ranges of intensities considered.
- With respect to a non-segmented structure, the implementation of inter-story isolation and realization of a non-conventional TMD produced a great attenuation of the dynamic structural response at all the frequencies, proving that it is a smart control strategy useful to enhance structural vibration mitigation.
- With respect to a three-story structure, the introduction of a TMD with a high mass ratio produced its control action mainly around the first mode, with a great attenuation of the dynamic response in a wide range of frequencies centered on the first uncontrolled mode.

**Author Contributions:** Conceptualization, M.B. and M.D.A.; methodology, M.D.A.; software, M.B.; validation, M.B. and M.D.A.; formal analysis, M.D.A.; investigation, M.B. and M.D.A.; resources, M.D.A.; data curation, M.B.; writing—original draft preparation, M.B. and M.D.A.; writing—review and editing, M.B. and M.D.A.; visualization, M.B.; supervision, M.D.A.; project administration, M.D.A.; funding acquisition, M.D.A. All authors have read and agreed to the published version of the manuscript.

**Funding:** This research received funding from Progetto Sapienza protocol number N. RP1181643697C751.

**Institutional Review Board Statement:** Not applicable.

**Informed Consent Statement:** Not applicable.

**Data Availability Statement:** Not applicable.

**Acknowledgments:** The authors would like to acknowledge the students of the course "Dynamics of Structures", held by Professor Maurizio De Angelis, for their active participation in the experimental activities.

**Conflicts of Interest:** The authors declare no conflict of interest.

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
