# Peer review of "Experimental Dynamic Response of a Multi-Story Frame Structure Equipped with Non-Conventional TMD Implemented via Inter-Story Isolation"

_applsci, doi:10.3390/app12189153_

Round 1

Reviewer 1 Report

The paper presents the results of a shaking table dynamic experimentation conducted on a 4-story steel frame structure equipped with non-conventional TMD implemented by inter-story isolation. The results can provide reference values for related engineering design. The topic of the paper is interesting and relevant for this journal. However there are still some issues with this article that need to be revised. It is recommended that the paper can be accepted.

1. The language is not clear and understandable enough and the sentences are always lengthy. For instance, “The original physical model is…576 x 580 mm” (Line100-104).

2. The abstract should be rewritten and important conclusions should be incorporated in the abstract.

3. The description of the experiment is very limited and inane.

4. The figures and tables in the paper are obscure, besides, the instructive texts in these figures and tables are also not clear enough. In Table 1, some arrows and instructive texts can be added for better illustrating the models.

5. In Line 153, “F3S the 3-story structure with the support plane for the HDRB”. Does the 3-story structure of the F3S configuration (Table 1) need to equip with the HDRB? It is not easy for readers to understand them clearly. Please explain it.

6. In Line 159, “NL=4-1 and NU=1 DOF”, can you please explain what does them mean? The two parameters do not shown in Fig.5a.

7. From Line 200-202, words are incompatible with Fig.7.

8. There are some format problems in the paper. For example, the font of Fig.17 is inconsistent with other graphs.

9. In the case of F4 configuration, the choice of value of the two parameters (i.e. the identified relevant frequencies; the identified damping factors) is different from the other two configurations, please explain why. This paper took a contrast between two of the models. Do you think it's reasonable?

10. The final conclusion is long. Please condense these conclusions, and it's better to divide into several concise conclusions.

Author Response

The authors thank the reviewer comments that allowed to improve the quality of the paper. All questions are in the following addressed and we hope to have clarified all the issues.

The modification in the revised manuscript are reported in red text. Moreover, notes have been added to relate the modifications to the reviews observations.

REVIEWER 1

Q1. The language is not clear and understandable enough and the sentences are always lengthy. For instance, “The original physical model is…576 x 580 mm” (Line100-104).

R1. According to the reviewer comment, the period indicated (Line100-104) has been shortened and rewritten. Moreover, when possible the sentences of the manuscript have been shortened.

Q2. The abstract should be rewritten and important conclusions should be incorporated in the abstract.

R2. Thanks to the reviewer comment, the abstract has been revised inserting relevant conclusions of the work, considered the word limit of 200 characters given by the Journal.

Q3. The description of the experiment is very limited and inane.

R3. The description of the experiment has been enriched and better commented in the revised manuscript. In Section 2 the physical models have been better described and the figures have been better commented and related to the text to describe adequately the models. In Section 4 the experimental set up and the inputs adopted are described, also this section has been updated. The description of the acquisition system has been added, the description of the actuator and the sensors utilized has been enriched and better related to the figures present in the manuscript.

Q4. The figures and tables in the paper are obscure, besides, the instructive texts in these figures and tables are also not clear enough. In Table 1, some arrows and instructive texts can be added for better illustrating the models.

R4. According to the reviewer observation, Table 1 has been enriched with text and arrows in order to explain better the models. Moreover, the most part of the figures present in the revised paper have been updated and increased quality.

Q5. In Line 153, “F3S — the 3-story structure with the support plane for the HDRB”. Does the 3-story structure of the F3S configuration (Table 1) need to equip with the HDRB? It is not easy for readers to understand them clearly. Please explain it.

R5. The sentence indicated has been rewritten in the revised manuscript in the following way: “F3S — the 3-story structure with the rigid support plane.” In fact, the F3S configuration is not equipped with the HDRB. Moreover, Table 1 has been modified and enriched to make more clear this configuration.

Q6. In Line 159, “NL=4-1 and NU=1 DOF”, can you please explain what does them mean? The two parameters do not shown in Fig.5a.

R6. The sentence has been modified in the revised text as: “The isolation system located below the 4th floor segments the 4-story frame structure into a lower (L) portion, also denominated as substructure, and an isolated upper (U) portion, also denominated as superstructure, corresponding to a number of DOF indicated as NL = 3 and NU = 1 DOF respectively, see in Fig. 5a.” NL and NU respectively represent the degrees of freedom of the lower structure, the substructure, and the upper structure, the superstructure. Fig. 5a has been modified to explain better the concept.

Q7. From Line 200-202, words are incompatible with Fig.7.

R7. The sentence indicated has been modified in the revised manuscript in the following way, to explain more clearly the position of the instruments in Fig. 7a: “One accelerometer measured the shaking table (AT) and the others the structural accelerations one at the first tree levels (A1, A2, A3), and two on the support plane and on the TMD mass respectively (AS1, AS2, A4.1, A4.2), Fig. 7a.”

Q8. There are some format problems in the paper. For example, the font of Fig.17 is inconsistent with other graphs.

R8. The font of Fig. 17 has been adjusted. Moreover, when possible, the format of other figures in the manuscript has been aligned.

Q9. In the case of F4 configuration, the choice of value of the two parameters (i.e. the identified relevant frequencies; the identified damping factors) is different from the other two configurations, please explain why. This paper took a contrast between two of the models. Do you think it's reasonable?

R9. The three adopted physical models, reported in Table 1, behave dynamically differently even if are built from a unique reference model (F4 configuration). So it is right expecting to obtain different identified parameters (frequencies and damping factors). However, we made for F4 configuration a new more refined identification, that we report in the revised paper, where a further mode (around 13 Hz, very difficult to identify) emerged. We updated the results that can be red in Section 5.1 in red color.

Q10. The final conclusion is long. Please condense these conclusions, and it's better to divide into several concise conclusions.

R10. According to the reviewer comment, the conclusions have been shortened and divided in comments in bullet points.

Reviewer 2 Report

There have been a lot of studies on the use of inter-story isolation for tuned mass contol. And there are a lot of relevant theoretical and experimental studies. There is no innovation in this paper, it is just the repetition of previous achievements.

Author Response

Q1. There have been a lot of studies on the use of inter-story isolation for tuned mass control. And there are a lot of relevant theoretical and experimental studies. There is no innovation in this paper, it is just the repetition of previous achievements.

R1. The authors are disappointed with the Reviewer opinion. However, they disagree with the observations made. Even if is true that inter-story isolation is a topic treated in literature, it is known that it is examined mainly concerning modelling aspects through theoretical studies. Experimental results on its effectiveness utilizing real physical models are few. The authors added two other literature studies specifically focusing on the experimentation:

  • Villaverde, R., Mosqueda, G. Aseismic roof isolation system: Analytic and shake table studies. Earthq. Eng. Struct. Dyn. 1999, 28(2-3), pp. 217-234.
  • Shi, S.; Du, D., Xu; W., Wang. S. Theoretical and experimental study on an innovative seismic retrofit solution for old brick masonry buildings. Eng. Struct. 2020, 225,111296.

 The papers report experiments but do not focus on the aspects explored in this paper, and treat different applications. For this reason, the authors think that the experimentation conducted within this study is meaningful. Moreover, compared with other previous experimentations on non-conventional TMD systems, in this study the following new original aspects are fully explored:

  • new experimentation on non-conventional TMD via interstory isolation that contemplates two reference models for comparisons: the non-segmented 4 story structure (F4 configuration), the 3-story structure (F3S configuration); it allowed to highlight experimentally the effect of floor segmentation and isolation via non-conventional TMD on an original 4-story frame structure, as well as the insertion of a TMD with large mass ratio as general control application on a three 3-story structure.
  • wide parametrical investigation on structural dynamics as well as control capacity, varying the type of excitation (white noise, increasing and decreasing sine sweep, natural far field and near fault earthquakes) and its intensity in a wide range of amplitudes.
  • Experimentation that evidences how a nonlinear non-conventional TMD system is robust and effective in the reduction of the dynamical response in a wide range of frequencies explored.

Reviewer 3 Report

Overall, very nice experimental study is presented that compare the isolation performance between the different configurations. The experimental design presented data and discussion is comprehensive in nature and is well written. Excellent work! A couple of minor points:

-   -The research gap could be articulated better at the end of the introduction section to ensure clarity.

-  - The authors chose a 4-story structure for their experiments. Could authors comment on how the non-conventional TMD may perform if the structure is even higher. I am not looking for any calculation or additional work but it would be interesting to know author’s perspective – which could be of interest to the readers.

Author Response

The authors thank the reviewer comments that allowed to improve the quality of the paper. All questions are in the following addressed and we hope to have clarified all the issues.

The modification in the revised manuscript are reported in red text. Moreover, notes have been added to relate the modifications to the reviews observations.

Q1. The research gap could be articulated better at the end of the introduction section to ensure clarity.

R1. Thanks to the reviewer observation, the introduction has been revised in order to better highlight the importance of the research and its original aspects compared to the current literature results. The reviewer can read Section1 Introduction.

Q2.  The authors chose a 4-story structure for their experiments. Could authors comment on how the non-conventional TMD may perform if the structure is even higher. I am not looking for any calculation or additional work but it would be interesting to know author’s perspective – which could be of interest to the readers.

R2. The results reported in the experimentation can be extended in general to a higher structure if inter-story isolation is implemented at the roof level or at the last upper stories. Thus, the application considered in this paper is oriented to medium rise buildings, of height typical in the Italian country. Differently if inter-story isolation is implemented by segmenting the high structure isolating a large number of floors, the deformability of the superstructure can play an important role and the model utilized in this paper, would not be valid for this case. In this latter case at least a 3DOF model should be adopted, where the further DOF represents the deformability of the superstructure.

Round 2

Reviewer 2 Report

The manuscript has been sufficiently improved.